# A breeding pool of ideas: Analyzing interdisciplinary collaborations at the Complex Systems Summer School

**Jacqueline Brown**[1]*, **Dakota Murray**[2], **Kyle Furlong**[3], **Emily Coco**[4], **Fabian Dablander**[5]

**1** Faculty of Environmental Studies, York University, Toronto, Canada, **2** School of Informatics, Computing, and Engineering, Indiana University Bloomington, Bloomington, Indiana, United States of America, **3** The MITRE Corporation, McLean, Virginia, United States of America, **4** Center for the Study of Human Origins, New York University, New York, New York, United States of America, **5** Department of Psychological Methods, University of Amsterdam, Amsterdam, Netherlands

* jackie.ilana.brown@gmail.com

**Data Availability Statement:** The data and code used in our analyses is available at https://github.com/fdabl/Analyzing-CSSS.

## Abstract

Interdisciplinary research is essential for the study of complex systems, and so there is a growing need to understand the factors that facilitate collaboration across diverse fields of inquiry. In this exploratory study, we examine the composition of self-organized project groups and the structure of collaboration networks at the Santa Fe Institute's Complex Systems Summer School. Using data from all iterations of the summer school from 2005 to 2019, comprising 823 participants and 322 projects, we investigate the factors that contribute to group composition. We first test for homophily with respect to individual-level attributes, finding that group composition is largely consistent with random mixing based on gender, career position, institutional prestige, and country of study. However, we find some evidence of homophilic preference in group composition based on disciplinary background. We then conduct analyses at the level of group projects, finding that project topics from the Social and Behavioral Sciences are over-represented. This could be due to a higher level of baseline interest in, or knowledge of, social and behavioral sciences, or the common application of methods from the natural sciences to problems in the social sciences. Consequently, future research should explore this discrepancy further and examine whether it can be mitigated through policies aimed at making topics in other disciplines more accessible or appealing for collaboration.

## 1 Introduction

"Among chosen combinations the most fertile will often be those formed of elements drawn from domains which are far apart."

- Henry [1].

**Funding:** The authors received no specific funding for this work.

**Competing interests:** The authors have declared that no competing interests exist.

"If you're going to do interdisciplinary studies and enter someone else's domain, the least you should do is take their questions very seriously. They've spent a long time formulating them."

- John Holland, as cited in [2].

Interdisciplinary research is essential for the study of complex phenomena, and so there is a growing need to understand the factors that facilitate collaboration across diverse fields of inquiry. Overspecialization has long been viewed as a root cause of fragmentation in science [3, 4], prompting many researchers to advocate for increased interdisciplinary work in order to break down silos and address problems spanning multiple subjects [5]. In light of its perceived benefits, many research institutions and funding agencies have developed programs to encourage interdisciplinary research and collaboration, such as the Canadian Institute for Advanced Research, the MacArthur Foundation, and the Santa Fe Institute [6].

There are, however, several barriers to interdisciplinary research. First, there are institutional challenges, including discipline-focused incentive structures around faculty hiring, promotion, and research funding [7, 8]. Second, there are cognitive obstacles stemming from conflicting epistemic values, conceptual frameworks, and methodologies [9, 10]. This can be particularly challenging in collaborations among natural and social scientists, especially when the contributions of the latter are devalued by the former. Social psychologist Thomas Heberlein argues that there is a widespread assumption that the natural sciences are more rigorous, and therefore more highly regarded, than the social sciences [11]. While empirical data on this phenomenon is limited, one study found that earth scientists consider social scientists to be significantly less competent than natural scientists [12]. However, the study also shows that earth scientists hold more positive perceptions of social scientists if they have worked with them in the past. This is in line with [13], who conclude that natural scientists who had previously collaborated with social scientists view themselves as better able to make intellectual contributions in an interdisciplinary setting than those who had not. It may be difficult to bridge this divide, however, without initiatives designed to build relationships between scholars from different disciplines. For example, few publications in groundwater research integrate both natural and social sciences, in spite of the field being inclined toward interdisciplinarity [14]. Furthermore, researchers in disciplines such as physics and chemistry are less likely to engage in interdisciplinary research than those affiliated with disciplines that are more strongly oriented to practical applications [15].

A deeper understanding of interdisciplinary research is becoming increasingly important with the growing recognition that "integrated studies of coupled human and natural systems reveal new and complex patterns and processes not evident when studied by social or natural scientists separately" [16]. Furthermore, greater computational power and access to large data sets have made it possible for researchers to pursue new scientific questions across disciplines [17]. As a result, there is a need for renewed study of interdisciplinary collaborations that employ advanced technological tools to address under-explored areas of inquiry.

In this paper, we conduct an exploratory study of interdisciplinary collaborations at the Santa Fe Institute's Complex Systems Summer School (CSSS). George Cowan, the Institute's founding president, felt that "the traditional disciplines had become so entrenched and so isolated from one another that they seemed to be strangling themselves" [2]. The Santa Fe Institute was conceived as an antidote to this problem. Today, it brings together scholars from a wide variety of disciplinary backgrounds to study complex adaptive systems and emergent phenomena, with CSSS serving as one of its leading educational programs. Described as "an

intensive four-week introduction to complex behavior in mathematical, physical, living, and social systems" [18], CSSS attracts and selects for a diverse set of graduate students, faculty, and professionals from around the world. Participants self-organize to undertake research projects on topics of their choosing, with many collaborations extending beyond the duration of the program.

Our study aims to understand what factors are associated with the composition of project groups at CSSS. We investigate individual and project-level characteristics. At the individual level, we explore participant gender, career position, institutional prestige, country of study, and disciplinary background as potential contributing factors. At the project level, we analyze the role of project discipline. Our research provides an opportunity to examine the structure of interdisciplinary collaborations that emerge in the absence of significant institutional constraints and presents a unique case study that may contribute to broader understandings of interdisciplinary research.

## 2 Methods

### 2.1 The Complex Systems Summer School

The Complex Systems Summer School (CSSS) is an annual month-long workshop hosted by the Santa Fe Institute in Santa Fe, New Mexico. The program advertises itself as broadly interdisciplinary, but with a focus on topics relevant to the study of complex systems, such as network theory, nonlinear dynamics, scaling, simulation, and statistical inference. Participants live and work together for the duration of the program, attend lectures and social events, and undertake one or more collaborative research projects that culminate in a presentation and paper.

Projects are developed entirely by participants, with topic selection and group formation occurring in several ways. Some participants suggest ideas and data sets from their own research, whereas others propose topics that interest them but are outside the scope of their usual work. In addition, many topics are inspired by lectures and conversations with professors and other participants throughout the program. In 2019, for example, a number of projects emerged from a brainstorming session held in the first week of the program, during which participants were encouraged to share ideas on a whiteboard and sign up for the topics they found most appealing. There are no limitations on project topics, nor are there any requirements regarding group composition. While projects with only one member are permitted, they rarely occur, accounting for only 17 of 322 total projects in our data. The modal group size of projects is 4. Participants must take part in at least one project, although they are typically involved in multiple.

The program solicits applications from graduate students, postdoctoral researchers, faculty, and government and industry professionals from around the world. The basic requirements include English proficiency and some STEM background, and in recent years, the program has encouraged applications from members of underrepresented groups and individuals from non-STEM disciplines. As of the 2019 iteration, the application consisted of two letters of recommendation, a CV, and a statement of research interests. There is no publicly available information on the evaluative criteria used to determine acceptance, although the program boasts a large and competitive pool of applicants. Upon admission, prospective participants are required to register and pay a fee, typically funded by their university, to cover program tuition, materials, accommodation, and meals. While not prohibited, our data does not include any participants who attended more than one iteration of the summer school.

## 2.2 Data collection

All data used in our analysis are publicly accessible. We manually scraped participant metadata from the publicly available CSSS Wiki for each iteration of the summer school held in Santa Fe between 2005 and 2019. These Wikis are used to aggregate logistic, lecture, and project information for the summer school, and facilitate communication between participants (an example Wiki can be found at https://wiki.santafe.edu/index.php/Complex_Systems_Summer_School_2019_(CSSS), and links to the other Wikis can be found at https://wiki.santafe.edu/index.php/Main_Page). The Wikis contain a biographical page for each participant, with self-reported personal details and academic interests. They also include project titles, descriptions, and group members. Our initial scraping of participant and project metadata yielded 1,024 participants and 363 projects. Unfortunately, the use of each Wiki was highly idiosyncratic with some variation in use over time. In particular, we excluded the 2011 iteration of CSSS from our analysis due to lack of biographical and disciplinary information. Similarly, we excluded individuals for whom no discipline could be ascertained from all discipline-based analyses. This issue primarily affected the 2005 iteration of CSSS, for which disciplinary information could not be established for the majority of participants. After accounting for these irregularities, we were left with 823 participants and 322 unique projects.

We extracted name, institutional affiliation, and biographical information for each participant listed on the Wiki for every available year. Institutional affiliations were manually cleaned and disambiguated. Position was categorized as "Student", "Faculty", which includes postdocs and professors, or "Not Academia", the latter of which were primarily affiliated with industry and government organizations. In addition, institutional prestige was assigned as a binary variable indicating whether the participant's institution was in the top 50 as rated by the 2019 Academic Rankings of World Universities [19]. A gender of "Male", "Female", or "Other/Unknown" was assigned to participants based on information they supplied in their biography. We manually assigned participant disciplines as one or more of the 19 categories that comprise the UNESCO ISCED Fields of Study [20] based on information supplied in their biography. In the case of multiple possible discipline classifications, a primary discipline was assigned based on a judgement among all the authors. If there was not enough information available to determine a discipline, we labeled it as unknown. For each project, we extracted the project title, description, and group members as stated on the Wiki or in the CSSS Proceedings Book, a collection of abstracts and associated authors for all projects in a given year. Based on the title and description, we manually assigned a UNESCO ISCED discipline to each project, as we did for participants. Anonymized data and code for our analysis is available from https://www.github.com/fdabl/Analyzing-CSSS/.

Descriptive statistics regarding each iteration of CSSS can be found in Table 1. While there is some year-to-year variation, participants were largely male and affiliated with academic institutions in the U.S. In addition, the majority were Master's or PhD students. The program has expanded in recent years, with 81 participants in 2019. Total counts of participants and projects by discipline over all years of CSSS can be found in Table 2.

## 2.3 Networks and measures

We assessed the structure of participant collaborations by constructing network representations for each iteration of the summer school (see S12–S25 Figs in S1 Appendix). A node was created for each participant (colored by participant discipline), with edges representing collaborations between participants on a project (colored by project discipline). All analyses were performed using R 3.6.1 [21], and network-based analyses were completed using the *DiagrammeR* package [22].

**Table 1. Descriptive data for the Complex Systems Summer School from 2005-2019.**

| Year | # Participants | # Projects | % Female | % U.S. | % Student | % Faculty | % Inter |
|------|---------------|-----------|----------|--------|-----------|-----------|---------|
| 2005 | 44 | 17 | 0.34 | 59.6 | - | - | 91.5 |
| 2006 | 45 | 22 | 0.42 | 58.5 | 67.9 | 7.5 | 47.2 |
| 2007 | 65 | 32 | 0.25 | 59.1 | 67.3 | 21.8 | 20.0 |
| 2008 | 53 | 17 | 0.24 | 47.0 | 77.3 | 12.1 | 18.2 |
| 2009 | 48 | 29 | 0.34 | 61.1 | 86.7 | 2.2 | 6.7 |
| 2010 | 56 | 22 | 0.29 | 52.1 | 82.3 | 6.2 | 8.3 |
| 2011 | - | - | - | - | - | - | - |
| 2012 | 48 | 19 | 0.22 | 58.7 | 84.1 | 14.3 | 23.8 |
| 2013 | 59 | 19 | 0.38 | 52.7 | 78.0 | 22.0 | 40.7 |
| 2014 | 54 | 22 | 0.32 | 38.8 | 76.7 | 19.8 | 44.0 |
| 2015 | 43 | 11 | 0.36 | 41.8 | 65.5 | 27.3 | 43.6 |
| 2016 | 79 | 24 | 0.33 | 53.8 | 72.4 | 15.2 | 18.6 |
| 2017 | 81 | 22 | 0.43 | 59.0 | 70.5 | 16.4 | 10.7 |
| 2018 | 67 | 23 | 0.46 | 59.8 | 75.5 | 17.6 | 21.6 |
| 2019 | 81 | 43 | 0.32 | 56.7 | 69.2 | 20.9 | 12.4 |

*% Inter.* refers to the proportion of participants working on a project outside of their own discipline.

Homophily is based on the principle, observed in many networks, that "contact between similar people occurs at a higher rate than among dissimilar people" [23]. We used a node-level homophily measure known as the Herfindahl Hirschman Index (HHI) to assess the degree to which individuals worked with a diverse set of participants based on group identity. HHI measures the concentration of a node's ego network in particular groups without

**Table 2. Counts of participants by discipline category, projects by discipline category, and mean, median, and modal project size.**

| Discipline | # Participants | # Projects | Mean Proj. Size | Median Proj. Size | Modal Proj. Size |
|------------|---------------|-----------|-----------------|-------------------|------------------|
| Life sciences | 146 | 62 | 3.9 | 4 | 2 |
| Social and behavioral sciences | 141 | 112 | 4.3 | 4 | 4 |
| Physical sciences | 136 | 11 | 4.5 | 4 | 4 |
| Computing | 86 | 45 | 4 | 4 | 2 |
| Engineering and engineering trades | 79 | 7 | 3.6 | 2 | 2 |
| Mathematics and statistics | 61 | 9 | 3.7 | 4 | 4 |
| Unknown | 57 | 4 | 3 | 2 | 2 |
| Health | 36 | 21 | 4.5 | 4 | 6 |
| Humanities | 28 | 13 | 4.7 | 5 | 5 |
| Business and administration | 18 | 9 | 5 | 4 | 4 |
| Environmental protection | 12 | 3 | 5.3 | 6 | 6 |
| Architecture and building | 10 | 10 | 6.1 | 5 | 10 |
| Arts | 7 | 7 | 5.7 | 6 | 3 |
| Agriculture and forestry and fishery | 2 | 3 | 3 | 3 | 1 |
| Journalism and information | 2 | 1 | 6 | 6 | 6 |
| Social services | 1 | 2 | 4.5 | 4 | 4 |
| Law | 1 | 0 | - | - | - |
| Teacher training and education science | 0 | 3 | 2.7 | 3 | 3 |

Discipline categories were assigned based on the UNESCO ISCED Fields of Study classifications. 2011 is excluded due to sparse and poor data.

considering the node's own attributes. This allows for more clearly assessing the diversity of a node's collaboration network without requiring the in-group and out-group dichotomy based on similarity involved in many other homophily measures. By using HHI, we can more directly address the question of whether participants are working with a diverse network of people, as opposed to whether they are collaborating with individuals similar to themselves. The latter is investigated via a dyadic analysis, which is explained below. HHI is calculated as the sum of the squares of the percentages of each group in a node's neighborhood [24]. In our study, a larger HHI value indicates that an individual worked primarily with people that share a particular trait, which implies that the node's ego network is more homophilous. Conversely, a smaller HHI value suggests a more equal distribution of connections among participants with different traits, indicating that the node's ego network is more heterophilous. HHI was calculated for every individual in the collaboration network and then aggregated by participant attribute for each year of CSSS.

In addition to homophily, we assessed the centrality of nodes in the network based on individual characteristics using two measures. The first measure, eigenvector centrality, or eigen-centrality, is an individual-focused measure that examines how well-connected a node is by weighting its centrality by the centrality of its connections [25, 26]. In our study, eigencentrality indicates the extent to which participants collaborated with individuals who were involved in many projects. To arrive at an eigencentrality score for a particular attribute (such as country or discipline), we averaged the score across individuals who have this attribute for each year. Such an individual-based score might inflate the centrality of individuals with a particular attribute, however. For example, if Social and Behavioral Sciences participants collaborate extensively, but only amongst themselves, then the average of their individual-based centrality score will be large and potentially misleading.

To account for variance among individual participants, we also measured the group degree centrality [27] for each participant attribute. Group degree centrality is calculated as the number of nodes outside of a group, as defined by a particular attribute, that are connected to members of that group. In order to compare across all years of CSSS, we normalized group degree centrality by the total number of participants not part of the group in question. We are primarily interested in assessing whether individuals are more likely to collaborate with participants outside their discipline that share a particular attribute. We report group centrality results for key disciplines in the main text and other centrality analyses in an appendix.

### 2.4 Individual-level analysis of networks

We adopted a null model simulation framework to test the extent to which the demographic attributes of participants relate to the composition of groups. We assume that if these factors have no effect, group composition should be consistent with random mixing. For each summer school collaboration network, we randomly shuffled the identities of nodes while maintaining edge connection, effectively randomizing groups. This process was repeated across 500 simulations. For each simulation trial, a network-level homophily score was calculated as the average HHI across all nodes for each demographic category. The actual network-level homophily score was compared against the distribution of simulated scores to assess whether group composition was random or if it was related to participant demographics. We conducted a similar null model simulation to test node position via eigencentrality and group degree centrality measures. Node attributes were randomly shuffled while maintaining edges, and centrality values were calculated and averaged across various node attributes. This process was repeated in 500 simulations. The actual averaged centrality values for each demographic

category were then compared to the distribution of centrality values from the null models to determine whether node position was associated with particular node attributes.

We also assessed the extent to which certain combinations of discipline-to-discipline collaborations (dyads) appeared, compared to what was expected under random mixing. Here, we define a dyad as a person-to-person connection, or an edge in the network, aggregated to their respective disciplines; for example, a group with 3 participants, one in Computing and two in Health, would contain three dyads: (Computing, Health), (Computing, Health), and (Health, Health). A greater than expected number of a given dyad could indicate preferential attachment between participants in certain disciplines. We repeat the null model procedure for each year of data, shuffling participant disciplines and calculating the mean number of each dyad across 500 simulations. For each year, we calculated the percent differences between the actual number of each dyad and the expected mean. An aggregate measure was calculated to reflect the tendency for participants from certain disciplinary pairs to work together, defined as the mean of the percent differences between the actual and expected number of dyads across all years.

### 2.5 Project-level analysis of networks

At the project level, we examined how the discipline of project topics contributed to the composition of groups. To do this, we first calculated the number of different disciplines per project group, normalized by the group size for all groups larger than one person. For individuals who were coded with multiple disciplines, only the individual's primary discipline was considered, and individuals without a discipline assignment were dropped from the analysis. Counts of unique disciplines were averaged over all of the years of CSSS for each project topic. We then compared the proportion of projects in each discipline to the proportion of participants from that discipline across all years of the summer school. This comparison allowed us to assess the degree to which the proportion of project topics in a particular field is correlated with the proportion of participants from that field.

## 3 Results

### 3.1 Individual-level results

Whereas there are disparities in who attends the Complex Systems Summer School (see Table 1), Fig 1a illustrates limited evidence that demographic factors are related to group composition at the summer school. Few observations are greater than one standard deviation away from the mean of the null distribution, and generally, these trends lack a clear and consistent pattern. For example, in 2007 and 2013, there was slightly less gender homophily than expected, but between 2013 and 2019, the degree of homophily was roughly consistent with the null model. Of the five factors examined, the clearest pattern of homophily was observed for participant discipline, for which seven years had higher HHI, or homophily, than expected under random mixing. Specifically, disciplinary HHI was greater than one standard deviation above the expected for 2012, 2013, 2016, and 2018. Moreover, in 2008, 2017, and 2019, HHI calculated on discipline was more than two standard deviations above the mean of the null distributions, the largest difference observed across all attributes and years.

We further examined the role of discipline in the composition of project groups by calculating the normalized group degree centrality for the five most common disciplines at the summer school (see Fig 1b). There were no clear patterns of consistently higher or lower than expected centrality for participants in Engineering, Life Sciences, or Physical Sciences. In 2007, Computing participants were less central than expected, though this is not consistent across other years. There was some evidence that participants in Social and Behavioral Sciences fields

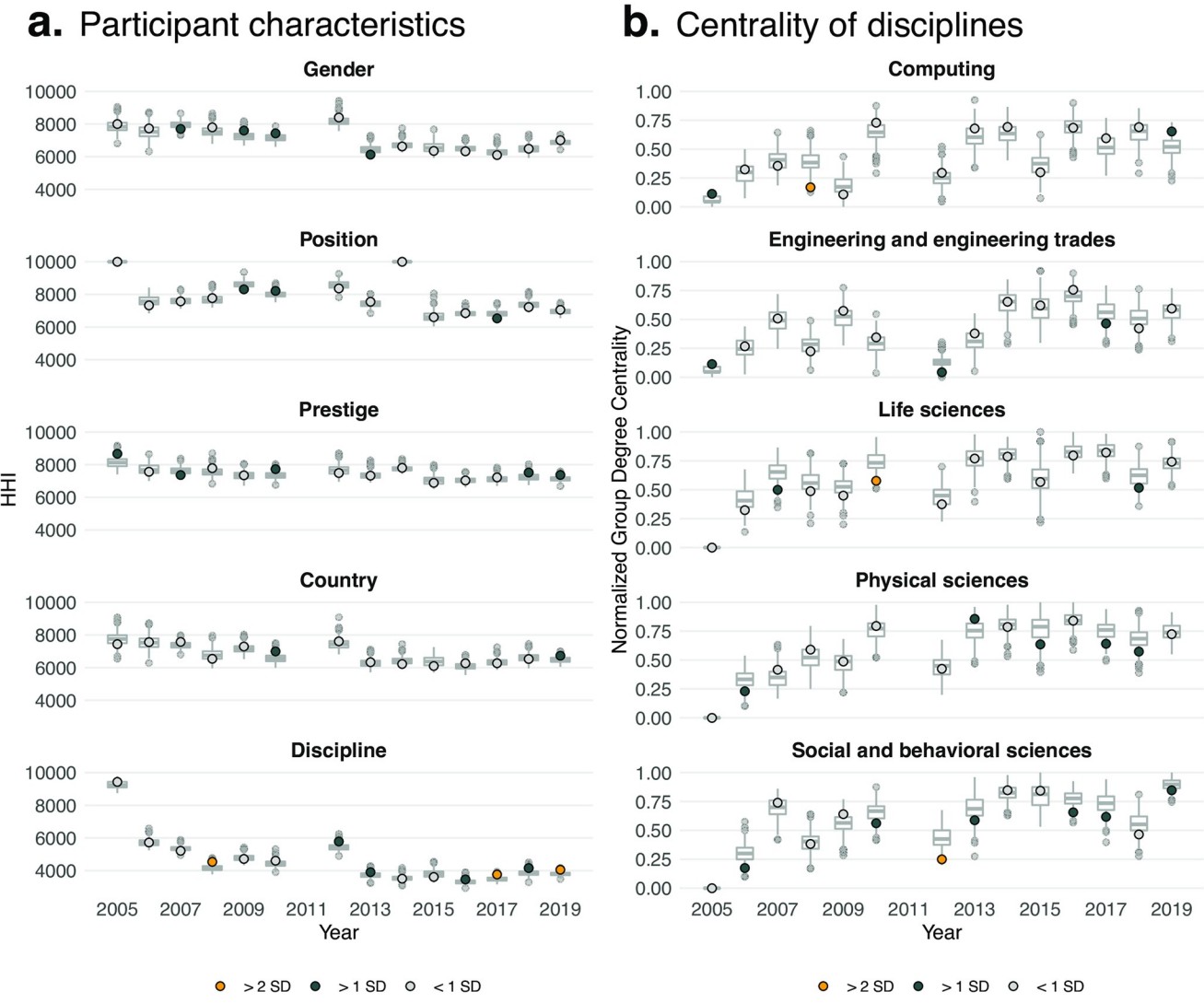

**Fig 1. a**. Group composition mostly consistent with random mixing. Grey boxplots correspond to the distribution of (**a**) network-level HHI values and (**b**) normalized group degree-centrality scores from 500 simulations of the null model. In each iteration of the simulation, participants are randomized while group links are maintained. Dots correspond to actual network-level metric for each summer school iteration and participant characteristic category. Dots are colored based on the number of standard deviations the actual value is away from the mean of the null distribution. **a**. Results are shown, over time, for gender, professional position, institutional prestige, country of study, and discipline. **b**. Results shown for the five disciplines with the most participants over the course of the summer school. Additional disciplinary results in appendix. 2011 is excluded due to sparse data.

were consistently less-central in the project networks than expected. In 2012, they were more than two standard below the expected centrality and more than one standard deviation below in 2006, 2010, 2013, 2016, 2017, and 2019.

We repeated this process for gender, position, institutional prestige, and country of study, and while we observe instances of higher and lower than expected group degree centrality compared to the null model in specific years, there were no clear or consistent patterns (see S1–S5 Figs in S1 Appendix). We also calculated the actual versus expected eigencentrality of participants in order to assess the relative influence of certain attributes in the network, but again observed no clear or consistent patterns (see S6–S10 Figs in S1 Appendix). These centrality measures may be influenced by other variables, however, such as individual personality

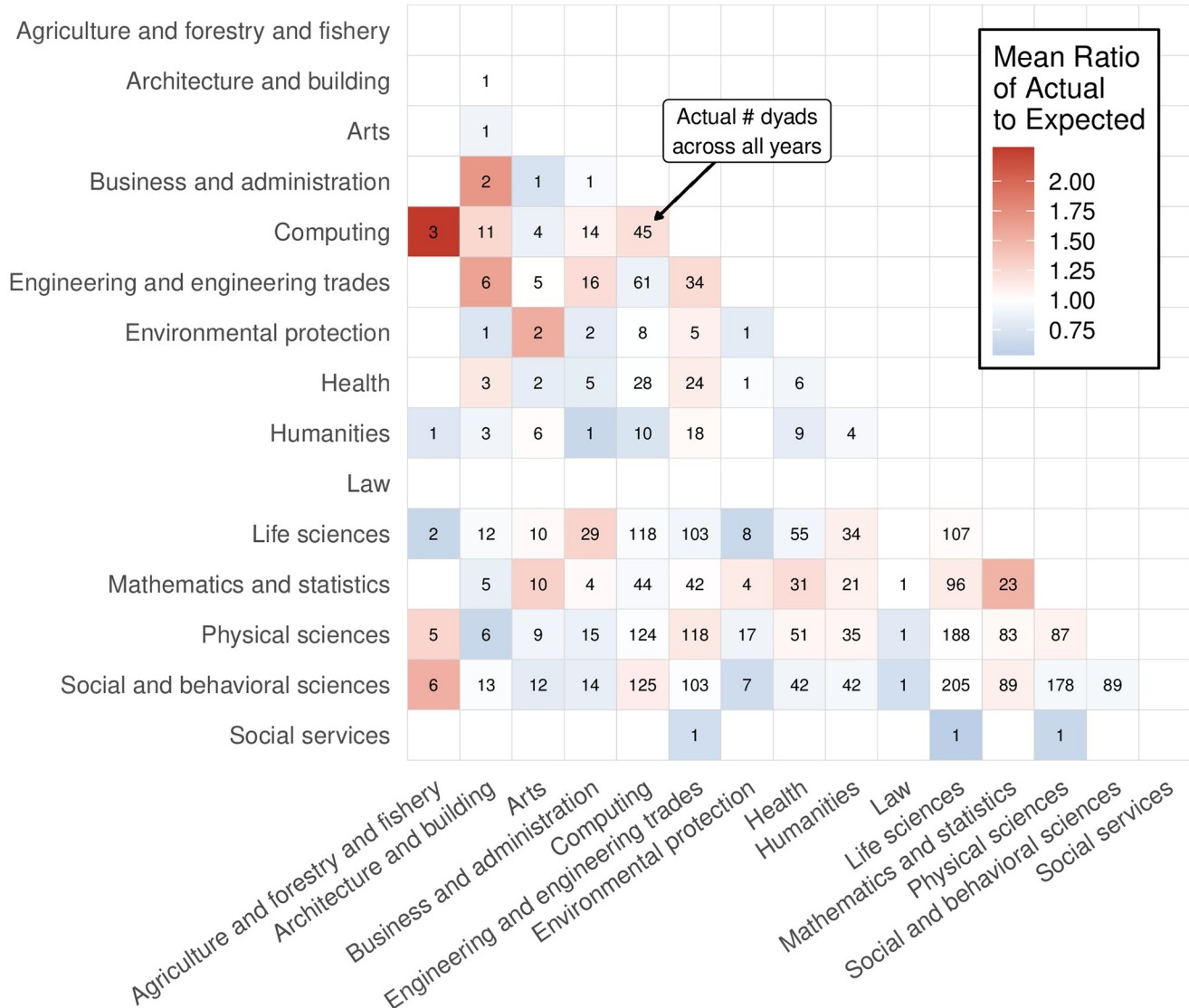

**Fig 2. Ratio of actual vs. expected number of each dyad across all iterations of the summer school excluding 2011 and 2005.** Color maps to the average of yearly ratio between the actual number of each dyads, and the expected value as calculated as the mean of number of dyads across 500 simulations of the null model. Red indicates a ratio greater than one, meaning that the dyad occurred more than expected; blue indicates a ratio less than one, meaning that the dyad occurred less than expected. The value in each cell corresponds to the count of each dyad across all years of the summer school.

traits and friendship ties. It is possible that the random patterns we find here result from one or more factors independent of the factors under study.

We further explored the role of discipline in group composition by examining the actual number of discipline-to-discipline connections (dyads) compared to the expected number of dyads based on the null model, broken down by summer school iteration (Fig 2). The most over-represented dyads with more than 20 participants were Mathematics and Statistics with themselves (1.56 times greater), Life Sciences with Business (1.29 times greater), and Engineering with themselves (1.22 times greater). In addition, dyads of traditionally quantitative disciplines tended to be somewhat over-represented, such as Computing with themselves (1.21 times greater), Physical Sciences and Engineering (1.15 times greater), and Physical Sciences and Mathematics and Statistics (1.05 times greater). The most under-represented pairs with

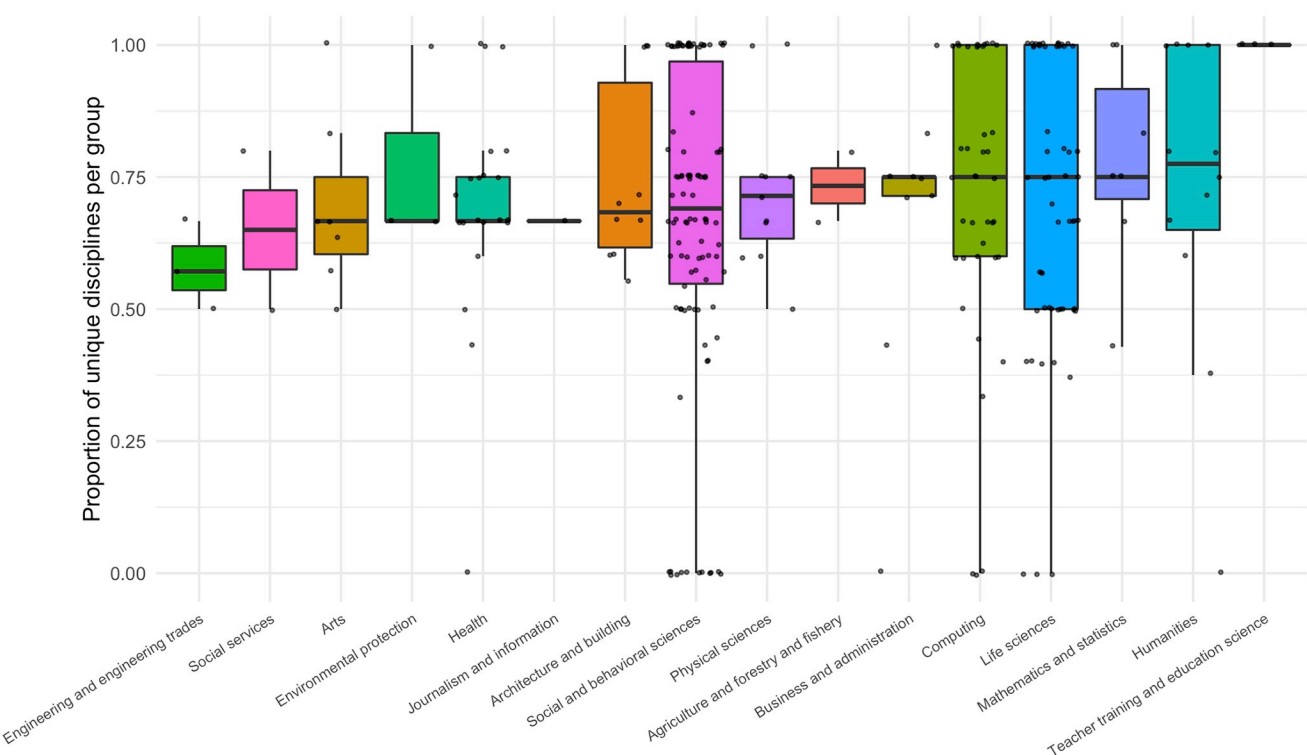

**Fig 3. Proportion of unique disciplines of participants within a project group averaged over all years of the Complex Systems Summer School summarized by topic of the project.** Values of 1 indicate that every member of a group is from a different discipline. Values of 0 mean that all group members are from the same discipline.

more than 20 participants were Computing and Engineering (0.88 times lower), Health and Life Sciences (0.9 times lower), and Life Sciences and Engineering (0.92 times lower). Whereas participants in traditionally-quantitative fields tended to work together, Computing is an exception, as participants in Computing were somewhat less likely than expected to work with those in Mathematics and Statistics (0.95 times lower) and somewhat more likely than expected to work with Social and Behavioral Sciences (1.12 times greater). Also of note is that participants in Social and Behavioral Sciences are somewhat less likely to work together than expected (0.94 times lower).

## 3.2 Project-level results

As shown in Fig 3, group diversity by project topic is high for all disciplines. This metric is somewhat sensitive to group size, considering that smaller groups are more likely to have extreme values (either completely homogeneous or completely heterogeneous group composition). Fields where the modal group size is only two, such as Life Sciences and Computing, may be especially sensitive (see Table 2). Despite this group size effect, there is an overall pattern of high diversity, which remains when only particular subsets of the data are considered, such as all groups comprising more than 2 people or group sizes that fall within the interquartile range (3 to 6 people, inclusively). In addition, even though there is variation in group diversity, all groups larger than one person are typically made up of participants from at least two unique disciplines. Accordingly, all project groups at CSSS tend to exhibit at least a nominal amount of interdisciplinarity.

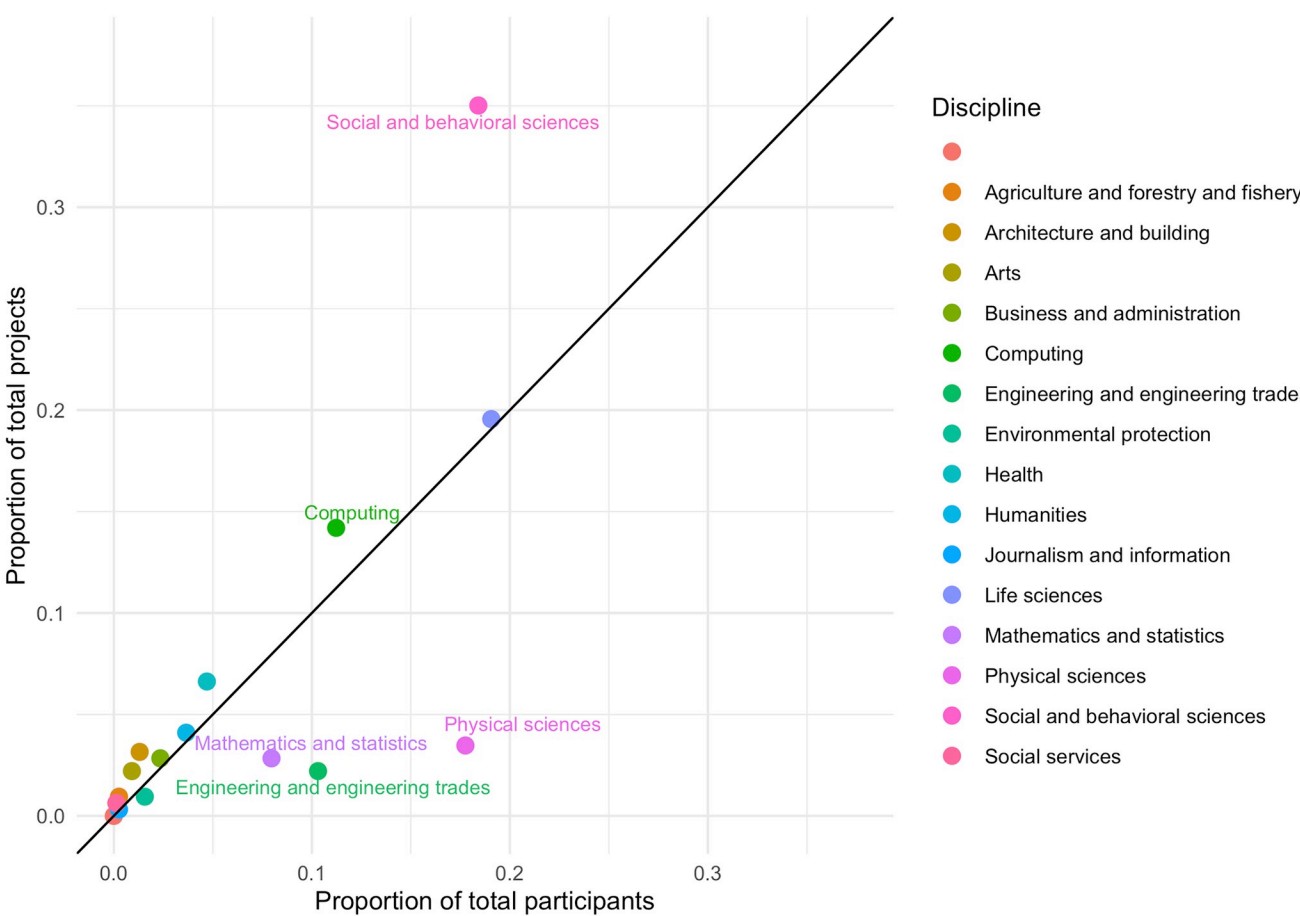

**Fig 4. Comparison of the proportion of participants from each discipline to the proportion of project topics in each discipline.** Law and Education are excluded because there were no projects and participants, respectively, from these disciplines as seen in Table 2. Points above the line have more projects than expected based on the number of participants; points below the line have fewer projects than expected. Labeled points had a difference in proportion greater than ±0.02.

For most disciplines, there tends to be a linear relationship between the proportion of participants from a specific discipline and the proportion of project topics from that same discipline, as shown in Fig 4. However, there are some outliers. For example, there are significantly more Social and Behavioral Sciences projects compared to the number of participants from this discipline. Conversely, there are far fewer projects in the Physical Sciences, Engineering, and Mathematics and Statistics compared to the number of participants from those disciplines. The pattern of Social and Behavioral Sciences projects being over-represented and Physical Sciences projects being underrepresented is consistent across most iterations of the summer school (see S11 Fig in S1 Appendix). In contrast, there was a greater degree of variation in the representation of Life Sciences and Computing projects.

## 4 Discussion

Our results suggest that project group composition at the Santa Fe Institute's Complex Systems Summer School does not follow patterns of academic collaboration found in many other studies. Rather than comprising like individuals, groups formed at CSSS more closely resemble a process of random mixing with respect to participant attributes such as gender, position, institutional prestige, and country of study. In particular, our individual-level analyses do not

demonstrate any homophilic preferences based on gender, in contrast to a growing body of literature indicating that researchers tend to collaborate with colleagues of the same gender [28–30], which, in male-dominated fields, can further exacerbate the marginalization of women in team science. While there have been more male attendees than female in every year of CSSS, with the disparity greater in some years than others, it appears that gender is not a significant factor in group composition once at the summer school. In a study of gendered differences in scientific labor, [31] observe that men are more likely to be associated with conceptual roles, whereas women are more likely to be associated with the "physical" labor of performing experiments. While beyond the scope of our study, it is possible that similar dynamics were present in collaborations at CSSS.

In part, the lack of homophily we observed may be due to the structure and goals of the program. For example, CSSS does not differentiate between participants of varying career stages, possibly mitigating traditional hierarchies [32]. Furthermore, unlike collaboration at many academic institutions, projects developed at CSSS are not tied to specific curricular or research expectations. Rather, they are intended to provide participants with opportunities to explore new ideas, regardless of whether those ideas will lead to publications or fulfill professional objectives.

The participant characteristic that appears to be most related to group composition is disciplinary background. Across seven years of the summer school, we observed slightly higher than expected disciplinary homophily on average. This finding was echoed by the dyad null models, which indicated that each of the six largest disciplines by number of participants displayed a larger than expected preference for working with individuals from their own discipline, with the notable exception of the Social and Behavioral Sciences. There was evidence for higher than expected preferential attachment between Social and Behavioral Sciences and three other disciplines. Disregarding one (Agriculture and Forestry and Fishery) due to small sample size, the two other disciplines were Computing and Mathematics and Statistics, which lends support to the argument that CSSS projects often combine tools and knowledge from traditionally quantitative and qualitative fields. While Social and Behavioral Sciences participants exhibited lower group degree centrality scores than expected across several years of the summer school, this is likely due to the fact that they were more likely to collaborate with the same people on multiple projects, and therefore had fewer unique connections than other large disciplines.

The results of the project-level analyses reveal more Social and Behavioral Sciences projects in comparison to the number of participants with this disciplinary background. This disparity suggests that these topics were heavily favored for interdisciplinary collaborations at the summer school. This is perhaps due to a higher level of baseline interest in or knowledge of topics related to human behavior and interaction. In contrast, there were far fewer projects in the Physical Sciences, Engineering, and Mathematics and Statistics in comparison to the number of participants from these disciplines, which indicates that these projects were less attractive for interdisciplinary collaboration at the summer school, perhaps due to the advanced technical concepts found in all three of these fields. Echoing the argument above, another possible explanation for the relatively low proportion of projects from the Physical Sciences, Engineering, and Mathematics and Statistics is that methods from these disciplines are being applied to projects in other areas. Techniques from statistical physics are increasingly being used to model social phenomena, such as crowd behavior, belief system dynamics, and the spread of disease [33]. Additionally, hybrid fields such as computational social science and digital humanities apply tools from mathematics and computing to traditionally qualitative fields. Interdisciplinary teams may be able to generate novel insights on a topic by applying

methodologies from other fields while still accounting for contextual factors supplied by those with subject matter expertise.

The trend of Social and Behavioral Sciences projects being over-represented and Physical Sciences projects being underrepresented was consistent across most years of the summer school (see S11 Fig in S1 Appendix). Conversely, there was a greater degree of variation in the representation of Life Sciences and Computing projects, which could mean that these fields have potential for high levels of interdisciplinary collaboration but are not favored in the same way as Social and Behavioral Sciences projects.

## 4.1 Limitations

Our study has several limitations. First, the disciplines of participants and projects were hand-coded by the authors. For some individuals, it was not clear to which UNESCO ISCED Field of Study they belonged, leading to potential ambiguities in disciplinary classification. Furthermore, CSSS attracts participants with interdisciplinary backgrounds, roughly 13% of whom were coded with more than one discipline. We decided to reduce each participant to their primary discipline to accommodate analysis; consequently, our results may not fully represent the disciplinary identity of an individual's area of study. Given the small number of participants coded with multiple disciplines, however, this should not significantly impact our findings. A similar problem existed for approximately 23% of projects, which were also interdisciplinary in nature, and for which we only considered the primary discipline in our analysis.

Second, our data is based on Wiki pages created and updated throughout each iteration of the summer school, which may not reflect final group members and topics at the time of project submission. We also did not investigate outcomes of projects developed at CSSS. These two limitations mean that our analysis may only represent an early snapshot of the dynamic collaboration networks during the summer school. Inferred collaboration networks could, therefore, reflect initial project interest rather than sustained collaborative effort, which may lead to an overestimation of group heterophily in some cases.

Third, it is likely that group composition is influenced by other demographic characteristics, or factors such as individual personality traits and friendship ties established throughout the program. For example, in a study of group formation among information systems undergraduate students, [34] find that people select others of the same race, as well as those who are reputed to be competent and dedicated workers. Due to a lack of relevant data, however, we were unable to account for these interpersonal attributes in our analyses.

Finally, the population analyzed in our study may not be representative of academic researchers more broadly. Our population was skewed toward U.S.-based researchers and those with STEM backgrounds. Complex systems is a field inclined toward interdisciplinarity; as such, CSSS participants are likely more inclined to pursue interdisciplinary research than other academics. The program's competitiveness and price tag may also dissuade individuals from less well-funded and less prestigious labs or universities from applying. Taken together, these factors limit the generalizability of our findings, and the outcomes of CSSS may not be replicable in other research contexts. However, we maintain that CSSS tears down traditional barriers and provides a unique case study of interdisciplinary research, which has value for understanding the dynamics of a novel educational program and perhaps for examining team science more broadly.

## 5 Conclusion

Interdisciplinary collaboration is essential to understanding a world of ever-increasing complexity. In this paper, we studied the factors influencing the emergence of such collaboration at

the Santa Fe Institute's Complex Systems Summer School. In this exploratory case study, we found that group composition is consistent with random mixing across several factors such as gender, institutional prestige, and academic discipline, a result that is heartening for an organization founded on the principle of tearing down disciplinary silos.

The over-representation of Social and Behavioral Sciences topics, however, is perhaps indicative of a unidirectional flow of knowledge in which individuals from other disciplines contribute heavily to projects in the Social and Behavioral Sciences such as economics, psychology, and sociology. It is possible that this work feeds back into other disciplines, with the rapid growth of computational social science exemplifying how computing tools have evolved as a result of modelling social and behavioral dynamics, but there are still topics related to engineering, mathematics, and physics that are disproportionately not pursue in interdisciplinary settings. Future research could investigate the impacts of this discrepancy on the production of scientific knowledge and explore how to make these topics more accessible for interdisciplinary research.

We provide a foundation for the analysis of collaboration networks formed during programs such as CSSS and also draw attention to gaps requiring further attention, such as how to encode multiple participant and project disciplines and how to effectively capture an evolving collaboration network. Other avenues of study include the analysis of publication metrics in order to better assess the program's overall impact and the incorporation of more robust disciplinary classifications that capture the *disciplinary distance* spanned by a collaboration.

Our study suggests that scaling up programs like the Complex Systems Summer School, which create the conditions for diverse collaboration networks to develop organically, may incentivize interdisciplinary research. As Poincaré put it, the most fertile combinations will indeed often be those drawn from domains which are far apart.

## Supporting information

**S1 Appendix.**
(PDF)

## Acknowledgments

We wish to thank the organizers of the Santa Fe Institute's Complex Systems Summer School, where this research originated, and Carrie Cowan for helpful comments on an earlier draft of this paper. We would also like to thank the two reviewers for their comments and suggestions that improved the quality of this manuscript.

Kyle Furlong's affiliation with The MITRE Corporation is provided for identification purposes only, and is not intended to convey or imply MITRE's concurrence with, or support for, the positions, opinions, or viewpoints expressed by the author. ©2020 The MITRE Corporation. ALL RIGHTS RESERVED. Approved for Public Release; Distribution Unlimited. Public Release Case Number 20-0611.

## Author Contributions

**Conceptualization:** Jacqueline Brown, Dakota Murray, Kyle Furlong, Emily Coco, Fabian Dablander.

**Data curation:** Jacqueline Brown, Dakota Murray, Kyle Furlong, Emily Coco, Fabian Dablander.

**Formal analysis:** Jacqueline Brown, Dakota Murray, Kyle Furlong, Emily Coco, Fabian Dablander.

**Investigation:** Jacqueline Brown, Dakota Murray, Kyle Furlong, Emily Coco, Fabian Dablander.

**Methodology:** Jacqueline Brown, Dakota Murray, Kyle Furlong, Emily Coco, Fabian Dablander.

**Software:** Dakota Murray, Kyle Furlong, Emily Coco, Fabian Dablander.

**Validation:** Jacqueline Brown, Dakota Murray, Kyle Furlong, Emily Coco, Fabian Dablander.

**Visualization:** Dakota Murray, Kyle Furlong, Emily Coco, Fabian Dablander.

**Writing – original draft:** Jacqueline Brown, Dakota Murray, Kyle Furlong, Emily Coco, Fabian Dablander.

**Writing – review & editing:** Jacqueline Brown, Dakota Murray, Kyle Furlong, Emily Coco, Fabian Dablander.

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
