## [Decision Letter · Decision Letter 0]

12 May 2020

PONE-D-20-08069

A Breeding Pool of Ideas: Analyzing Interdisciplinary Collaborations at the Complex Systems Summer School

PLOS ONE

Dear Ms. Brown,

Thank you for submitting your manuscript to PLOS ONE. After careful consideration, we feel that it has merit but does not fully meet PLOS ONE’s publication criteria as it currently stands. Therefore, we invite you to submit a revised version of the manuscript that addresses the points raised during the review process.

Your manuscript presents a potentially interesting case study on multidisciplinarity. However, as you can see below, some very relevant aspects of your work need a throughout revision. Such a revision should pay special attention to the following points:

- Embeddedness of your work in the previously existing literature. What gaps are you addressing here? To what extend your findings can contribute to the research of other scholars in similar topics?

- Presentation of the case study. Somehow related to the previous point (and also to the following one), it is really important to provide enough contextual details about the CS school in general and the process of group formation in particular.

- Methodology. All the applied methodology must be properly justified and explain (notice PLOS publication criterion #3, https://journals.plos.org/plosone/s/criteria-for-publication#loc-3). Comment number 5 by Reviewer 1 is a specific example, but Reviewer 2 points out other ones.

- Interpretation of results. Conclusions reached must be well based on findings (see PLOS publication criterion #4, https://journals.plos.org/plosone/s/criteria-for-publication#loc-4). In particular, none of the reviewers was satisfied with your claims about hierarchies.

We would appreciate receiving your revised manuscript by Jun 26 2020 11:59PM. To enhance the reproducibility of your results, we recommend that if applicable you deposit your laboratory protocols in protocols.io, where a protocol can be assigned its own identifier (DOI) such that it can be cited independently in the future. For instructions see: http://journals.plos.org/plosone/s/submission-guidelines#loc-laboratory-protocols

We look forward to receiving your revised manuscript.

Kind regards,

Sergi Lozano

Academic Editor

PLOS ONE

Journal Requirements:

Reviewers' comments:

Reviewer's Responses to Questions

**Comments to the Author**

1. Is the manuscript technically sound, and do the data support the conclusions?

Reviewer #1: Partly

Reviewer #2: No

2. Has the statistical analysis been performed appropriately and rigorously? 

Reviewer #1: Yes

Reviewer #2: No

3. Have the authors made all data underlying the findings in their manuscript fully available?

Reviewer #1: Yes

Reviewer #2: Yes

4. Is the manuscript presented in an intelligible fashion and written in standard English?

Reviewer #1: Yes

Reviewer #2: Yes

5. Review Comments to the Author

Reviewer #1: This paper represents a potentially useful addition to knowledge on interdisciplinarity, providing something of a unique angle, namely, observing what type of and to what extent interdisciplinary relations form amongst a diverse group of researchers when institutional constraints are not in play. The authors provide a large amount of data - compared to many interdisciplinary studies - and some means for analyzing the degree of interdisciplinary engagement across projects. The conclusions should help provoke some reflection by those studying interdisciplinarity and the presumption that disciplines do not interact necessarily freely or equally.

The study is relatively minimalistic with respect to its goals. This is of course a data-driven approach. However I believe more information should be provided to support the ability of readers to properly situate these results and assess their generalizability. The authors do state they think this data has relevance for understanding how interdisciplinarity might occur in practice once free of institutional constraints. So my points here are with that agenda in mind.

1. In the first place we should know more about the CSSS. One concern is that such a summer school provides certain structures and incentives which heavily favour interdisciplinary engagements, or even require them, within a generally low-stakes environment. Such a study would not necessarily allow us to draw inferences about what might happen in a "real research context" free of institutional constraints.

2. I did not find mentioned anywhere - apologies if I missed it - what the distribution of participants was amongst different levels (grad students, postdoc, professors etc). This is important given the prominent presumption that early career academics or students are much more open to ID and less parochial in their disciplinary affiliations than senior academics. Hence the results again show more flexible interaction than might otherwise be observed. And one might presume that a summer school tends to be dominated by those at an early stage.

3. I would also be worried that studying a summer school on "complexity science" already somewhat distorts the outcomes, on the basis that it is already a field orientated towards interdisciplinarity, and peopled by researchers with more open-mindedness about ID than your average researcher. Some accounting for this possibility would seem important.

4. It wasn't clear to me how project topics ended up with a disciplinary classification or where these topics come from.

5. In the study influence is used as a proxy for ID hierarchies, via an eigencentrality measure. This needs some clarification; firstly how influence is derived from eigencentrality, and secondly to what extent influence captures fully what is meant by hierarchy with respect to interdisciplinarity. Measuring connections between disciplines does not necessarily itself tell us about "influence", unless I misunderstand things. A discipline might be widely applied in many projects, but still occupy a somewhat minimal (not particularly influential) position in how a project is run, whose questions dominate, how the problem is framed and so on. Such a discipline presumably would have low overall influence, and I have seen business scientists and other social scientists take on many roles on many projects across the university landscape, but usually in a kind of service role with little control over those projects.

6. I would note to this last point, as a speculative aside really, that the fact that many collaborations occur on social science topics but not physical topics might be taken as evidence of hierarchies playing a role, to the extent that natural science researchers feel competence and authority to apply their knowledge and methods to social science topics, which is not felt the other way around. This would suggest some deep level presumptions about the relative value of natural (and engineering) science vs social science.

Reviewer #2: Review « A Breeding Pool of Ideas: Analyzing Interdisciplinary Collaborations at the Complex System Summer School »

The authors present a study of the group formation process in the Complex System Summer Schools from 2002 to 2019. They study if gender, position (in Academia or not), affiliation, country of study, and discipline can be relevant factors to predict the composition of a working group. They found that they are not since the composition is compatible with a random mixing. Such an a priori unexpected results compared to the literature should be properly discussed and investigated further but it is not. The following presented results by the authors remain obscure to me, especially the reason why they are presented.

One of my main points is that authors show that, for the possible factor they consider, the group formation composition is compatible with a random mixing. This means not only that there is no more homophily than expected from the participants once year, but also that there is no more heterophily, and thus no multidisciplinarity which would have been created intentionally. In other word, we can also say that the discipline was not considered as a criterion for the creation of the group. Thus, the authors can’t say they study the factors influencing group formation as it is written in the abstract since finally they just say they don’t know what influence group formation. Thus, this is necessary that the authors explain their global methodology since this is not clear to me why they do so many investigations to show what they have already shown by comparing their group composition to neutral models.

This leads me to my second point. Since the authors de facto observe multidisplinarity due to the simple initial co-presence of several disciplinarity in annual groups of participants, one would expect that they qualify this observed multidisciplinarity, especially indicating which couple of disciplines are preferentially linked to each other. This is particularly what we expect when they arrive to study the composition of groups in terms of disciplines. This would be particular relevant for their results regarding social and behavioral sciences one the one hand, and mathematics, statistics, physics, and engineering on the other hand. Can it be explained by the fact the latter are more coupled to social and behavioral scientists than the other disciplines. This would be relevant at this moment to discuss the results in comparison of (Barthel & Seidl, 2017).

A third point is about their study of hierarchy between disciplines. This is really not convincing. Indeed, their criteria for hierarchy are not convincing. Indeed, such they conclude the group have not been formed depending on disciplines, the simple fact that some participants is highly connected to other participants can have so many interpretations: they are more agreeable, they have a discipline which is more related with other disciplines for applied work, for example statistics, they are taller than others, they took the same bus to come to the summer school, … Thus finally I don’t know what the hierarchy they’re looking at refers to, especially I’m not sure that it refers to social values given to their different disciplines by researchers.

My last point is about the conclusion and the discussion. The authors begin the discussion telling that they study the factors influencing the emergence of such collaboration in an arguably ideal setting. “Ideal” seems to be very arbitrary and I would be pleased if they have explained at the beginning of the paper why their conditions for studying emergence of collaboration is ideal compared to other studies. Overall the discussion is poor; especially on their main result which is they found no relationship between the factors they consider and the group compositions. Some references lacks, for example the one of (Hinds, Carley, Krackhardt, & Wholey, 2000) related to the homophily in the choice of work group member. Moreover, discussing your study in light of the works made by (Darbellay, 2015; Sedooka, Steffen, Paulsen, & Darbellay, 2015) on interdisciplinarity and the identity of researchers doing interdisciplinarity, especially the fact that most of them have already many disciplines, should be interesting.

Complementary comments:

What is a group is not clearly defined. Indeed it appeared that a participant can be a member of different groups one year, while authors talk about groups composed of one participant.

Especially the HHI measure is not properly explained in the context of the particular network the authors used. Especially, does it mean they use two networks: one for group project and the other for discipline homophily? The economical explanation of the indicator does not speak to me much.

Also, regarding “testing homophily”, what the authors have done is not clear. Indeed, sometimes they talk about every disciplines (for example in the presented network in supplementary information and much of the presented graphs), and other times as in 2.3 I understand that they gather disciplines in two categories “social science” and “physical or natural science”. Thus what their results about the absence of homophily in the group composition mean is finally not clear to me. Is it only homophily in these two large categories?

References

Barthel, Roland, & Seidl, Roman. (2017). Interdisciplinary Collaboration between Natural and Social Sciences – Status and Trends Exemplified in Groundwater Research. PLOS ONE, 12(1), e0170754. doi: 10.1371/journal.pone.0170754

Darbellay, Frédéric. (2015). Rethinking inter- and transdisciplinarity: Undisciplined knowledge and the emergence of a new thought style. Futures, 65, 163-174. doi: https://doi.org/10.1016/j.futures.2014.10.009

Hinds, Pamela J., Carley, Kathleen M., Krackhardt, David, & Wholey, Doug. (2000). Choosing Work Group Members: Balancing Similarity, Competence, and Familiarity. Organizational Behavior and Human Decision Processes, 81(2), 226-251. doi: https://doi.org/10.1006/obhd.1999.2875

Sedooka, Ayuko, Steffen, Gabriela, Paulsen, Theres, & Darbellay, Frédéric. (2015). Paradoxe identitaire et interdisciplinarité : un regard sur les identités disciplinaires des chercheurs. [Identity Paradox and Interdisciplinarity: An Analysis of the Disciplinary Identities of Researchers]. Natures Sciences Sociétés, 23(4), 367-377. doi: 10.1051/nss/2015056

6. PLOS authors have the option to publish the peer review history of their article (what does this mean?). If published, this will include your full peer review and any attached files.

Reviewer #1: No

Reviewer #2: No

---

## [Author Response · Author response to Decision Letter 0]

3 Aug 2020

July 28, 2020

Dear Dr. Lozano and Reviewers,

Thank you for providing such detailed feedback. We have taken it seriously and substantially revised our manuscript. In particular, we have rewritten the introduction and discussion to better embed our results in the literature, we have added more detail about the Complex Systems Summer School, we have restructured our methodology so as to more clearly distinguish individual-level from project-level analyses, and we have added new analyses. Please find our response to each point below.

Reviewer #1:

1. In the first place we should know more about the CSSS. One concern is that such a summer school provides certain structures and incentives which heavily favour interdisciplinary engagements, or even require them, within a generally low-stakes environment. Such a study would not necessarily allow us to draw inferences about what might happen in a "real research context" free of institutional constraints.

This is a good point, and we added the following paragraph to Section 2.1 (Methods - The Complex Systems Summer School) in order to provide more detail about CSSS and the ways in which project groups are formed:

“Projects are developed entirely by participants, with topic selection and group formation occurring in several ways. Some participants suggest ideas and data sets from their own research, whereas others propose topics that interest them but are outside the scope of their usual work. In addition, many topics are inspired by lectures and conversations with professors and other participants throughout the program. In 2019, for example, a number of projects emerged from a brainstorming session held in the first week of the program, during which participants were encour- aged to share ideas on a whiteboard and sign up for the topics they found most appealing. There are no limitations on project topics, nor are there any requirements regarding group composition. While projects with only one member are permitted, they rarely occur, accounting for only 17 of 321 total projects in our data. The modal group size of projects is 4. Participants must take part in at least one project, although they are typically involved in multiple.”

We also added a table (Table 2) that contains the number of projects and number of participants in each discipline, in order to provide the reader with further information about our data set.

2. I did not find mentioned anywhere - apologies if I missed it - what the distribution of participants was amongst different levels (grad students, postdoc, professors etc). This is important given the prominent presumption that early career academics or students are much more open to ID and less parochial in their disciplinary affiliations than senior academics. Hence the results again show more flexible interaction than might otherwise be observed. And one might presume that a summer school tends to be dominated by those at an early stage.

Indeed, this distribution was not included in the previous draft. We have therefore updated Table 1 to include the distribution of students, postdocs, and professors across each year of CSSS. We also added the following statement to Section 2.2 (Methods - Data Collection) to reflect that the prevalence of students and postdocs in the program may lead to more flexible interaction:

“While there is some year-to-year variation, participants were largely male and affiliated with academic institutions in the U.S. In addition, the majority were Master’s or PhD students.”

3. I would also be worried that studying a summer school on "complexity science" already somewhat distorts the outcomes, on the basis that it is already a field orientated towards interdisciplinarity, and peopled by researchers with more open-mindedness about ID than your average researcher. Some accounting for this possibility would seem important.

Yes, this does seem likely. It is difficult to account for this in our paper, however, as it is a case study of exactly such a potentially more open-minded population. We have added some cautionary words to our limitations (Section 4.1) to reflect this point:

“Complex systems is a field inclined toward interdisciplinarity; as such, CSSS participants are likely more inclined to pursue interdisciplinary research than other academics.”

4. It wasn't clear to me how project topics ended up with a disciplinary classification or where these topics come from.

To add more clarity, we modified Section 2.2 (Methods - Data Collection) to better describe how project topics were classified with disciplines:

“For each project, we extracted the project title, description, and group members as stated on the Wiki or in the CSSS Proceedings Book, a collection of abstracts and associated authors for all projects in a given year. Based on the title and description, we manually assigned a UNESCO ISCED discipline to each project, as we did for participants.”

The paragraph we added in response to the first comment by Reviewer #1 offers an explanation as to how project topics were developed.

5. In the study influence is used as a proxy for ID hierarchies, via an eigencentrality measure. This needs some clarification; firstly how influence is derived from eigencentrality, and secondly to what extent influence captures fully what is meant by hierarchy with respect to interdisciplinarity. Measuring connections between disciplines does not necessarily itself tell us about "influence", unless I misunderstand things. A discipline might be widely applied in many projects, but still occupy a somewhat minimal (not particularly influential) position in how a project is run, whose questions dominate, how the problem is framed and so on. Such a discipline presumably would have low overall influence, and I have seen business scientists and other social scientists take on many roles on many projects across the university landscape, but usually in a kind of service role with little control over those projects.

We have removed the term “influence” as well as all references to a hierarchy of science. Instead, we shifted the focus of our interpretation to group composition. While we agree that there may be instances where tools from a particular discipline are widely applied in many projects yet still play a minor role, we unfortunately cannot assess this in our case study. We have added a paragraph to our limitations (Section 4.1) to clarify that our network results may be influenced by factors that we were unable to study:

“Third, it is likely that group composition is influenced by other demographic characteristics, or factors such as individual personality traits and friendship ties established throughout the program. For example, in a study of group formation among information systems undergraduate students, Hinds et al. [34] find that people select others of the same race, as well as those who are reputed to be competent and dedicated workers. Due to a lack of relevant data, however, we were unable to account for these interpersonal attributes in our analyses.”

While this is a limitation, we still feel that an analysis using such centrality measures is insightful. In particular, the null models we now use allow us to state that eigencentrality is not related to discipline, gender, position, and institutional prestige. It is instead likely driven by unobserved personality factors; yet the point that they are not related to the variables we studied such as discipline remains.

We have further supplemented our initial use of eigencentrality with group degree centrality, because eigencentrality is an individual-based metric that when aggregated can lead to inflated scores. In Section 2.3 (Methods - Networks and Measures), we write:

“In addition to homophily, we assessed the centrality of nodes in the network based on individual characteristics using two measures. The first measure, eigenvector centrality, or eigencentrality, is an individual-focused measure that examines how well-connected a node is by weighting its centrality by the centrality of its connections [25, 26]. In our study, eigencentrality indicates the extent to which participants collaborated with individuals who were involved in many projects. To arrive at an eigencentrality score for a particular attribute (such as country or discipline), we averaged the score across individuals who have this attribute for each year. Such an individual- based score might inflate the centrality of individuals with a particular attribute, however. For example, if Social and Behavioral Sciences participants collaborate extensively, but only amongst themselves, then the average of their individual-based centrality score will be large and potentially misleading.

To account for variance among individual participants, we also measured the group degree centrality [27] for each participant attribute. Group degree centrality is calculated as the number of nodes outside of a group, as defined by a particular attribute, that are connected to members of that group. In order to compare across all years of CSSS, we normalized group degree centrality by the total number of participants not part of the group in question. We are primarily interested in assessing whether individuals are more likely to collaborate with participants outside their discipline that share a particular attribute. We report group centrality results for key disciplines in the main text and other centrality analyses in an appendix.”

Overall, we have substantially revised our methodology and we hope that the changes sufficiently address the reviewer’s point.

6. I would note to this last point, as a speculative aside really, that the fact that many collaborations occur on social science topics but not physical topics might be taken as evidence of hierarchies playing a role, to the extent that natural science researchers feel competence and authority to apply their knowledge and methods to social science topics, which is not felt the other way around. This would suggest some deep level presumptions about the relative value of natural (and engineering) science vs social science.

This is an interesting point, which we touch on in our Introduction, Discussion, and Conclusion. In particular, in Section 5 (Conclusion), we write:

“The over-representation of Social and Behavioral Sciences topics, however, is perhaps indicative of a unidirectional flow of knowledge in which individuals from other disciplines contribute heavily to projects in the Social and Behavioral Sciences such as economics, psychology, and sociology. It is possible that this work feeds back into other disciplines, with the rapid growth of computational social science exemplifying how computing tools have evolved as a result of modelling social and behavioral dynamics, but there are still topics related to engineering, mathematics, and physics that are disproportionately not pursue in interdisciplinary settings. Future research could investigate the impacts of this discrepancy on the production of scientific knowledge and explore how to make these topics more accessible for interdisciplinary research.”

Reviewer #2:

1. The authors present a study of the group formation process in the Complex System Summer Schools from 2002 to 2019. They study if gender, position (in Academia or not), affiliation, country of study, and discipline can be relevant factors to predict the composition of a working group. They found that they are not since the composition is compatible with a random mixing. Such an a priori unexpected results compared to the literature should be properly discussed and investigated further but it is not. The following presented results by the authors remain obscure to me, especially the reason why they are presented.

We expanded on our discussion (Section 4) in order to compare our results to existing literature. In particular, we contrasted our findings with previous work on gender homophily in academic collaborations. We also reorganized the presentation of our results substantially in order to provide increased clarity; see our comment to the next point.

2. One of my main points is that authors show that, for the possible factor they consider, the group formation composition is compatible with a random mixing. This means not only that there is no more homophily than expected from the participants once a year, but also that there is no more heterophily, and thus no multidisciplinarity which would have been created intentionally. In other words, we can also say that the discipline was not considered as a criterion for the creation of the group. Thus, the authors can’t say they study the factors influencing group formation as it is written in the abstract since finally they just say they don’t know what influences group formation. Thus, this is necessary that the authors explain their global methodology since this is not clear to me why they do so many investigations to show what they have already shown by comparing their group composition to neutral models.

We revised the framing of our analyses to indicate that we examined the factors that impact group composition at two levels: individual-level and project-level. Four of the five attributes we chose to examine at the individual level (gender, career position, university prestige, and country of study) appear to be consistent with random mixing. Some of our new analyses, however, do indicate greater than expected homophily with respect to discipline (see response to next point). In addition, the project-level analyses indicate that the discipline of the project may have been a factor in group composition. We have further changed our framing from “influencing group formation” to “group composition”, as, strictly speaking, the former is a dynamic process for which we only observe its outcome (the composition of groups), not the process itself (what influences group formation).

3. This leads me to my second point. Since the authors de facto observe multidisciplinarity due to the simple initial co-presence of several disciplinarity in annual groups of participants, one would expect that they qualify this observed multidisciplinarity, especially indicating which couple of disciplines are preferentially linked to each other. This is particularly what we expect when they arrive to study the composition of groups in terms of disciplines. This would be particularly relevant for their results regarding social and behavioral sciences one the one hand, and mathematics, statistics, physics, and engineering on the other hand. Can it be explained by the fact the latter are more coupled to social and behavioral scientists than the other disciplines. This would be relevant at this moment to discuss the results in comparison of (Barthel & Seidl, 2017).

This is a great suggestion. We integrated a new analysis into our paper to assess whether any particular pairs of disciplines (dyads) exhibit preferential attachment for working together (see Figure 2). Here is the description of the method used, which can be found in Section 2.4 (Methods - Individual-Level Analysis of Networks):

“We also assessed the extent to which certain combinations of discipline-to-discipline collabora- tions (dyads) appeared, compared to what was expected under random mixing. Here, we define a dyad as a person-to-person connection, or an edge in the network, aggregated to their respective disciplines; for example, a group with 3 participants, one in Computing and two in Health, would contain three dyads: (Computing, Health), (Computing, Health), and (Health, Health). A greater than expected number of a given dyad could indicate preferential attachment between participants in certain disciplines. We repeat the null model procedure for each year of data, shuffling par- ticipant disciplines and calculating the mean number of each dyad across 500 simulations. For each year, we calculated the percent differences between the actual number of each dyad and the expected mean. An aggregate measure was calculated to reflect the tendency for participants from certain disciplinary pairs to work together, defined as the mean of the percent differences between the actual and expected number of dyads across all years.”

We report the outcomes of this analysis in Section 3.1 (Results - Individual-Level Results)

“We further explored the role of discipline in group composition by examining the actual number of discipline-to-discipline connections (dyads) compared to the expected number of dyads based on the null model, broken down by summer school iteration (Figure 2). The most over-represented dyads with more than 20 participants were Mathematics and Statistics with themselves (1.56 times greater), Life Sciences with Business (1.29 times greater), and Engineering with themselves (1.22 times greater). In addition, dyads of traditionally quantitative disciplines tended to be somewhat over-represented, such as Computing with themselves (1.21 times greater), Physical Sciences and Engineering (1.15 times greater), and Physical Sciences and Mathematics and Statistics (1.05 times greater). The most under-represented pairs with more than 20 participants were Computing and Engineering (0.88 times lower), Health and Life Sciences (0.9 times lower), and Life Sciences and Engineering (0.92 times lower). Whereas participants in traditionally-quantitative fields tended to work together, Computing is an exception, as participants in Computing were somewhat less likely than expected to work with those in Mathematics and Statistics (0.95 times lower) and somewhat more likely than expected to work with Social and Behavioral Sciences (1.12 times greater). Also of note is that participants in Social and Behavioral Sciences are somewhat less likely to work together than expected (0.94 times lower).”

Ultimately, the results of the dyad analysis lend further support to our argument regarding the over-representation of Social and Behavioral Sciences projects and under-representation of projects from traditionally quantitative fields. We have added this to our discussion (Section 4):

“This finding was echoed by the dyad null models, which indicated that each of the six largest disciplines by number of participants displayed a larger than expected preference for working with individuals from their own discipline, with the notable exception of the Social and Behavioral Sciences. There was evidence for higher than expected preferential attachment between Social and Behavioral Sciences and three other disciplines. Disregarding one (Agriculture and Forestry and Fishery) due to small sample size, the two other disciplines were Computing and Mathematics and Statistics, which lends support to the argument that CSSS projects often combine tools and knowledge from traditionally quantitative and qualitative fields.”

4. A third point is about their study of hierarchy between disciplines. This is really not convincing. Indeed, their criteria for hierarchy are not convincing. Indeed, such they conclude the group have not been formed depending on disciplines, the simple fact that some participants is highly connected to other participants can have so many interpretations: they are more agreeable, they have a discipline which is more related with other disciplines for applied work, for example statistics, they are taller than others, they took the same bus to come to the summer school, … Thus finally I don’t know what the hierarchy they’re looking at refers to, especially I’m not sure that it refers to social values given to their different disciplines by researchers.

We agree with this point, and have removed all references to a hierarchy of science. We also added the following statement to our limitations (Section 4.1) to indicate that group composition may have been influenced by factors that we were unable to measure:

“Third, it is likely that group composition is influenced by other demographic characteristics, or factors such as individual personality traits and friendship ties established throughout the program. For example, in a study of group formation among information systems undergraduate students, Hinds et al. [34] find that people select others of the same race, as well as those who are reputed to be competent and dedicated workers. Due to a lack of relevant data, however, we were unable to account for these interpersonal attributes in our analyses.”

5. My last point is about the conclusion and the discussion. The authors begin the discussion telling that they study the factors influencing the emergence of such collaboration in an arguably ideal setting. “Ideal” seems to be very arbitrary and I would be pleased if they have explained at the beginning of the paper why their conditions for studying emergence of collaboration is ideal compared to other studies. Overall the discussion is poor; especially on their main result which is they found no relationship between the factors they consider and the group compositions. Some references lacks, for example the one of (Hinds, Carley, Krackhardt, & Wholey, 2000) related to the homophily in the choice of work group member. Moreover, discussing your study in light of the works made by (Darbellay, 2015; Sedooka, Steffen, Paulsen, & Darbellay, 2015) on interdisciplinarity and the identity of researchers doing interdisciplinarity, especially the fact that most of them have already many disciplines, should be interesting.

We have rewritten our introduction and discussion substantially, and we have also removed references to CSSS as an “ideal” setting, instead noting that it is a unique case study. Our updated discussion includes references to many of the works mentioned above, as well as to other publications with relevance to group composition in academic settings.

6. What is a group is not clearly defined. Indeed it appeared that a participant can be a member of different groups one year, while authors talk about groups composed of one participant.

Especially the HHI measure is not properly explained in the context of the particular network the authors used. Especially, does it mean they use two networks: one for group projects and the other for discipline homophily? The economical explanation of the indicator does not speak to me much.

The paragraph we added in response to the first comment by Reviewer #1 adds clarity as to how project topics were developed, with the following sentences in particular referencing groups composed of one participant:

“There are no limitations on project topics, nor are there any requirements regarding group composition. While projects with only one member are permitted, they rarely occur, accounting for only 17 of 321 total projects in our data. The modal group size of projects is 4. Participants must take part in at least one project, although they are typically involved in multiple.”

We also removed the percent similarity homophily indicator and updated our explanation of the HHI measure in Section 2.3 (Methods - Networks and Measures):

“Homophily is based on the principle, observed in many networks, that “contact between similar people occurs at a higher rate than among dissimilar people” [23, p. 416]. We used a node-level homophily measure known as the Herfindahl Hirschman Index (HHI) to assess the degree to which individuals worked with participants with characteristics similar to their own. HHI measures the concentration of a node’s ego network in particular groups without considering the node’s own attributes. It is calculated as the sum of the squares of the percentages of each group in a node’s neighborhood [24]. In our study, a larger HHI value indicates that an individual worked primarily with people that share a particular trait, which implies that the node’s ego network is more homophilous. Conversely, a smaller HHI value suggests a more equal distribution of connections among participants with different traits, indicating that the node’s ego network is more heterophilous. HHI was calculated for every individual in the collaboration network and then aggregated by participant attribute for each year of CSSS.”

7. Also, regarding “testing homophily”, what the authors have done is not clear. Indeed, sometimes they talk about every discipline (for example in the presented network in supplementary information and much of the presented graphs), and other times as in 2.3 I understand that they gather disciplines in two categories “social science” and “physical or natural science”. Thus what their results about the absence of homophily in the group composition mean is finally not clear to me. Is it only homophily in these two large categories?

Whereas previously, our null models used the binary classification of “social science” and “physical or natural science” for discipline, we have updated this to look instead at all disciplines in our data set. The results of the new simulation (Figure 1. a) display some evidence for disciplinary homophily.

We hope that we have addressed each point satisfactorily.

Sincerely,

Jacqueline Brown, Dakota Murray, Kyle Furlong, Emily Coco, and Fabian Dablander

---

## [Decision Letter · Decision Letter 1]

24 Dec 2020

PONE-D-20-08069R1

A Breeding Pool of Ideas: Analyzing Interdisciplinary Collaborations at the Complex Systems Summer School

PLOS ONE

Dear Dr. Brown,

First of all, I would like to apologise for the delayed decision.

Thank you for submitting your manuscript to PLOS ONE. After careful consideration, we feel that it has merit but does not fully meet PLOS ONE’s publication criteria as it currently stands. Therefore, we invite you to submit a revised version of the manuscript that addresses the points raised during the review process.

You have correctly responded to all the comments and concerns by the two reviewers and, consequently, Reviewer 1 has recommended publication of your manuscript. Nevertheless, before proceeding, I would like you to address the following issues:

1.-You are using the Herfindahl Hirschman Index (HHI) as a measure of homophily. Could you, please, elaborate on the convenience of using this index instead of, for instance, assortativity?

2.- The boxplot in Fig3 shows that some disciplines (e.g. Social and behavioural sciences) present a significant amount of values equal to one (so, all groups composed by a single discipline). Have you checked whether this is related to smaller group sizes? As you normalized 'discipline diversity' by group size (to be able to compare across cases) this cannot be checked directly.

3.- In page 7, first paragraph, the text reads "Specifically, disciplinary HHI was above one standard deviation above the expected for 2012, 2013, 2016, and 2018." Maybe this sentence could be re-written to avoid using 'above' twice.

We look forward to receiving your revised manuscript.

Kind regards,

Sergi Lozano

Academic Editor

PLOS ONE

Reviewers' comments:

Reviewer's Responses to Questions

**Comments to the Author**

1. If the authors have adequately addressed your comments raised in a previous round of review and you feel that this manuscript is now acceptable for publication, you may indicate that here to bypass the “Comments to the Author” section, enter your conflict of interest statement in the “Confidential to Editor” section, and submit your "Accept" recommendation.

Reviewer #1: All comments have been addressed

2. Is the manuscript technically sound, and do the data support the conclusions?

Reviewer #1: Yes

3. Has the statistical analysis been performed appropriately and rigorously? 

Reviewer #1: Yes

4. Have the authors made all data underlying the findings in their manuscript fully available?

Reviewer #1: Yes

5. Is the manuscript presented in an intelligible fashion and written in standard English?

Reviewer #1: Yes

6. Review Comments to the Author

Reviewer #1: The authors have in my view adequately addressed my specific concerns. I am happy for the paper to be accepted with further revision.

7. PLOS authors have the option to publish the peer review history of their article (what does this mean?). If published, this will include your full peer review and any attached files.

Reviewer #1: **Yes: **Miles MacLeod

---

## [Author Response · Author response to Decision Letter 1]

11 Jan 2021

Dear Dr. Lozano and Reviewers,

Thank you for your feedback. We have incorporated your comments into the most recent draft of our manuscript. Please find our response to each point below:

1. You are using the Herfindahl Hirschman Index (HHI) as a measure of homophily. Could you, please, elaborate on the convenience of using this index instead of, for instance, assortativity?

We chose to use HHI instead of an index such as assortativity because it allows us to examine the diversity of a participant’s collaborations without considering the attributes of that particular individual. Given that not all groups are equally represented in our dataset, using a metric based on similarity between ego and the nodes in its network could potentially bias our measures of homophily of collaborations. For example, for an individual in a rare discipline like Social Services, a similarity-based homophily measure would necessarily indicate a more heterogeneous set of collaborations simply because there are no other Social Services participants to work with. It is possible, however, that Social Services participants exclusively work with individuals from the Social and Behavioral Sciences, meaning that their collaboration networks are actually not diverse. Using HHI allows us to see this information. Later on in the paper, we use a dyad analysis to examine the tendency of individuals to work with others from their discipline; if we had used a similarity-based homophily measure, we would be doubly analyzing this phenomenon. We have added language summarizing these points in Section 2.3:

“HHI measures the concentration of a node's ego network in particular groups without considering the node's own attributes. This allows for more clearly assessing the diversity of a node's collaboration network without requiring the in-group and out-group dichotomy based on similarity involved in many other homophily measures. By using HHI, we can more directly address the question of whether participants are working with a diverse network of people, as opposed to whether they are collaborating with individuals similar to themselves. The latter is investigated via a dyadic analysis, which is explained below.”

2. The boxplot in Fig3 shows that some disciplines (e.g. Social and behavioural sciences) present a significant amount of values equal to one (so, all groups composed by a single discipline). Have you checked whether this is related to smaller group sizes? As you normalized 'discipline diversity' by group size (to be able to compare across cases) this cannot be checked directly.

This is a good question. We looked into it and found that for all disciplines in which completely homogenous groups occur, the median and mode size of these groups is 2. However, we still found an overall pattern of diversity in group composition, which we tested first by excluding groups of 2 from the data, and then by considering only groups that fall within the interquartile range (3 to 6 individuals, inclusively). We have revised Section 3.2 accordingly:

“This metric is somewhat sensitive to group size, considering that smaller groups are more likely to have extreme values (either completely homogeneous or completely heterogeneous group composition). Fields where the modal group size is only two, such as Life Sciences and Computing, may be especially sensitive (see Table~\\ref{tab:Disc-Table}). Despite this group size effect, there is an overall pattern of high diversity, which remains when only particular subsets of the data are considered, such as all groups comprising more than 2 people or group sizes that fall within the interquartile range (3 to 6 people, inclusively).”

We have also updated the caption of Fig3 to add clarity:

“Proportion of unique disciplines of participants within a project group averaged over all years of the Complex Systems Summer School summarized by topic of the project. Values of 1 indicate that every member of a group is from a different discipline. Values of 0 mean that all group members are from the same discipline.”

In addition, we have updated Table 2 to include the mean, median, and modal group size for each discipline.

3. In page 7, first paragraph, the text reads "Specifically, disciplinary HHI was above one standard deviation above the expected for 2012, 2013, 2016, and 2018." Maybe this sentence could be re-written to avoid using 'above' twice.

We have re-written this sentence to say the following:

“Specifically, disciplinary HHI was greater than one standard deviation above the expected for 2012, 2013, 2016, and 2018.”

Finally, we have made minor tweaks to the manuscript, as well as identifying and fixing an error in our data processing, leading to 823 rather than 824 participants, and 322 rather than 321 projects. We have also updated the figures that relied on this data.

We hope that we have addressed each point satisfactorily.

Sincerely,

Jacqueline Brown, Dakota Murray, Kyle Furlong, Emily Coco, and Fabian Dablander

---

## [Editor Report · Decision Letter 2]

18 Jan 2021

A Breeding Pool of Ideas: Analyzing Interdisciplinary Collaborations at the Complex Systems Summer School

PONE-D-20-08069R2

Dear Dr. Brown,

We’re pleased to inform you that your manuscript has been judged scientifically suitable for publication and will be formally accepted for publication once it meets all outstanding technical requirements.

Kind regards,

Sergi Lozano

Academic Editor

PLOS ONE
---

## [Editor Report · Acceptance letter]

21 Jan 2021

PONE-D-20-08069R2 

A Breeding Pool of Ideas: Analyzing Interdisciplinary Collaborations at the Complex Systems Summer School 

Dear Dr. Brown:

I'm pleased to inform you that your manuscript has been deemed suitable for publication in PLOS ONE. Congratulations! Your manuscript is now with our production department. 

Kind regards, 

on behalf of

Dr. Sergi Lozano 

Academic Editor

PLOS ONE